# In Vitro Activity of Selected Phenolic Compounds against Planktonic and Biofilm Cells of Food-Contaminating Yeasts

**DOI:** 10.3390/foods10071652

**Published:** 2021-07-17

**Authors:** Bernard Gitura Kimani, Erika Beáta Kerekes, Csilla Szebenyi, Judit Krisch, Csaba Vágvölgyi, Tamás Papp, Miklós Takó

**Affiliations:** 1Department of Microbiology, Faculty of Science and Informatics, University of Szeged, Közép Fasor 52, H-6726 Szeged, Hungary; loyaltykings2012@gmail.com (B.G.K.); kerekeserika88@gmail.com (E.B.K.); szebecsilla@gmail.com (C.S.); csaba@bio.u-szeged.hu (C.V.); pappt@bio.u-szeged.hu (T.P.); 2MTA-SZTE “Lendület” Fungal Pathogenicity Mechanisms Research Group, University of Szeged, Közép Fasor 52, H-6726 Szeged, Hungary; 3Institute of Food Engineering, Faculty of Engineering, University of Szeged, Mars tér 7, H-6724 Szeged, Hungary; krisch@mk.u-szeged.hu

**Keywords:** antimicrobial activity, biofilm, antiadhesion, spoilage yeasts, natural phenolics

## Abstract

Phenolic compounds are natural substances that can be obtained from plants. Many of them are potent growth inhibitors of foodborne pathogenic microorganisms, however, phenolic activities against spoilage yeasts are rarely studied. In this study, planktonic and biofilm growth, and the adhesion capacity of *Pichia anomala*, *Saccharomyces cerevisiae*, *Schizosaccharomyces pombe* and *Debaryomyces hansenii* spoilage yeasts were investigated in the presence of hydroxybenzoic acid, hydroxycinnamic acid, stilbene, flavonoid and phenolic aldehyde compounds. The results showed significant anti-yeast properties for many phenolics. Among the tested molecules, cinnamic acid and vanillin exhibited the highest antimicrobial activity with minimum inhibitory concentration (MIC) values from 500 µg/mL to 2 mg/mL. Quercetin, (−)-epicatechin, resveratrol, 4-hydroxybenzaldehyde, *p*-coumaric acid and ferulic acid were also efficient growth inhibitors for certain yeasts with a MIC of 2 mg/mL. The *D. hansenii*, *P. anomala* and *S. pombe* biofilms were the most sensitive to the phenolics, while the *S. cerevisiae* biofilm was quite resistant against the activity of the compounds. Fluorescence microscopy revealed disrupted biofilm matrix on glass surfaces in the presence of certain phenolics. Highest antiadhesion activity was registered for cinnamic acid with inhibition effects between 48% and 91%. The active phenolics can be natural interventions against food-contaminating yeasts in future preservative developments.

## 1. Introduction

Food spoilage refers to the process of food quality deterioration affecting the nutritional and sensorial properties of food [1]. Microbiological food spoilage is a major concern nowadays due to the economic losses caused [2]. Yeasts are renowned as great fermenters of alcoholic and non-alcoholic beverages, have a key role in bread making, and are producers of products of high interest, such as volatile aroma compounds or biofuels [3]. Despite the positive application of yeasts in food production, spoilage yeasts may lead to food deterioration and loss, which is a common problem in the food industry [4,5]. Yeasts, such as *Saccharomyces cerevisiae*, *Debaryomyces hansenii*, *Zygosaccharomyces rouxii*, *Pichia anomala*, *Schizosaccharomyces pombe* and *Zygosaccharomyces bailii* have been mentioned not only in food production but also in food spoilage [5,6,7,8,9]. Yeast contamination of foods often leads to biofilm formation, obnoxious flavors, generation of off odors and other unsavory effects [8].

Many microorganisms including yeasts can closely associate with a multitude of surfaces in their natural environment to form biofilms. Adhesion of cells to the surface is an important step in the colonization and biofilm formation, thereby in the development of mature biofilm forms [10]. Mature biofilms are multicellular aggregates composed of proteins, polysaccharides and nucleic acid enveloped in a self-generated biopolymer matrix [11]. In such a community, microorganisms have lower growth rates and behave differently from the planktonic cells [12]. Biofilms have unique survival properties such as increased resistance to antimicrobial agents, enhanced genetic exchanges, and increased synthesis of secondary metabolites [13,14]. In the last decades, there has been a growing interest in studying microbial biofilm formation due to its implications in the medical field, food processing and the environment [14,15].

In recent years, natural compounds as preservatives have gained attention in the food industry because of the consumers’ growing concern against chemical additives. Chemical additives are thought to be synthetic materials and the main concern is that their long-term use could lead to cancers or other disorders [16]. The immediate reactions by some sensitive individuals against colorants or other additives may include hypersensitivity, allergy or headache [17].

Phenolic compounds are secondary metabolites in plants known for their health-promoting and antimicrobial properties. Plant phenolics have different groups, such as phenolic acids, flavonoids, stilbenes, lignans and tannins, which are defined according to the number of phenol rings that the compound contain and the nature of structural elements between the rings [18,19]. Excellent sources of phenolic compounds are the antioxidative fruits and vegetables, but many herbs and spices can also contain a large amount of them [20,21]. Extraction of phenolics from plant residues can be performed through physical, chemical, fermentation and enzyme-assisted approaches [22,23]. The bioactive phenolics can be used as natural preservatives since they are safer and more ecofriendly than synthetic agents due to their natural origin. In this context, several studies dealt with the antimicrobial and antibiofilm properties of phenolic compounds [24,25,26,27], mainly presenting data on the effects against bacteria. Experiments have been performed with some spoilage yeasts as well. For instance, vanillin and grape phenolics such as pterostilbene, resveratrol, luteolin, *p*-coumaric acid and ferulic acid were investigated against the planktonic growth of *S. cerevisiae*, *Z. bailii*, *Z. rouxii* and other wine-related yeasts [28,29,30]. Phenolics from essential oils and foods, e.g., coumarins, curcumin and pyrogallol, as well as various plant polyphenolic extracts have been shown to be effective against yeast biofilms [9,19]. However, antimicrobial properties of many phenolic compounds, especially their effect on adhesion, are still unexplored in this group of microorganisms.

In this study, the potential antimicrobial, antibiofilm and antiadhesion properties of phenolic compounds belonging to different phenolic groups, i.e., hydroxybenzoic acids, hydroxycinnamic acids, stilbenes, flavonoids and phenolic aldehydes, were investigated against four spoilage yeasts. This work provides useful data for the potential application of natural phenolic compounds as antimicrobial agents against spoilage yeasts.

## 2. Materials and Methods

### 2.1. Yeast Strains and Growth Conditions

Four yeasts, i.e., *P. anomala* SZMC 8061Mo, *S. cerevisiae* SZMC 1279, *S. pombe* SZMC 1280 and *D. hansenii* SZMC 8045Mo, were involved to the assays. All yeast strains were obtained from the Szeged Microbiological Collection (SZMC) maintained by the Department of Microbiology of the University of Szeged, Szeged, Hungary (http://www.wfcc.info/ccinfo/collection/by_id/987, accessed on 21 June 2021). *P. anomala*, *S. cerevisiae* and *S. pombe* were grown on malt extract medium containing 5 g/L yeast extract (Biolab, Budapest, Hungary), 5 g/L glucose (Biolab, Budapest, Hungary) and 50 mL/L 20% (*v*/*v*) malt extract (Merck, Budapest, Hungary). *D. hansenii* was cultivated on yeast extract peptone dextrose (YPD) medium containing 4 g/L glucose (Biolab, Budapest, Hungary), 4 g/L peptone (Sigma–Aldrich, Munich, Germany) and 2 g/L yeast extract (Biolab, Budapest, Hungary). Before each assay, fresh yeast cultures were prepared through incubation for 24 h at 30 °C. At the end of the incubation period, growth of each yeast was in the stationary phase. After preparing a serial dilution with the corresponding growth medium, cell number was set by counting them in a Bürker chamber under a light microscope.

### 2.2. Phenolic Compounds

Phenolic compounds from different classes of phenolics were involved to the assay (Table 1). These phenolics were (i) hydroxybenzoates, i.e., gallic acid, vanillic acid, syringic acid, 4-hydroxybenzoic acid and protocatechuic acid; (ii) hydroxycinnamates, i.e., cinnamic acid, ferulic acid, *p*-coumaric acid and caffeic acid; (iii) stilbenes, i.e., resveratrol and polydatin; (iv) flavonoids, i.e., (−)-epicatechin and quercetin; and (v) phenolic aldehydes, i.e., vanillin and 4-hydroxybenzaldehyde. The phenolic compounds were purchased from Sigma–Aldrich (Munich, Germany). A stock solution with a concentration of 4 mg/mL was prepared from each phenolic compound in 10% (*v*/*v*) ethanol. Most phenolics included in the tests were selected based on our previous investigations. Namely, they were identified in grape, apple and pitahaya residue samples after enzyme extraction and showed antimicrobial and antibiofilm activities against food-related bacteria [27,31]. The other compounds tested proved to be effective antimicrobial agents in other studies [28,32,33].

### 2.3. Growth Inhibition Activity and Calculation of Minimum Inhibitory Concentrations (MICs)

The inhibitory effect of phenolic compounds against the planktonic growth of yeasts was assayed through a microplate method [27]. Briefly, stock solutions of each phenolic compound were serially diluted with 10% (*v*/*v*) ethanol to the concentrations from 31.25 µg/mL to 2 mg/mL. A volume of 100 µL from the diluted samples and from the corresponding stock solution were transferred to the wells of a 96-well polystyrene microtiter plate (Sarstedt, Nümbrecht, Germany). Then, 100 µL of cell suspension (10^5^ CFU/mL) prepared in double concentrated medium was added to each well, giving a final concentration from 15.625 to 2 mg/mL for the compounds during the test. Positive controls contained the inoculated growth medium without any phenolic compound, while the negative controls had the phenolic inhibitors in a sterile medium. After 24 h incubation at 30 °C, absorbance was measured at 600 nm using a SPECTROstar Nano (BMG Labtech, Offenburg, Germany) microplate reader. The concentration of the phenolic compound that caused 90% or higher growth inhibition, i.e., where the absorbance was 10% or lower than that of the positive control was considered as the MIC. All measurements were carried out in three biological and three technical parallels.

### 2.4. Biofilm Formation and Treatment

The effect of phenolic compounds on the biofilm formation was examined by following the method of Zambrano et al. [27]. Wells of a 96-well polystyrene microtiter plate (Sarstedt, Nümbrecht, Germany) were filled with 200 µL of 24 h old yeast culture with approximately 10^8^ CFU/mL concentration. After 4 h of cell adhesion performed at 30 °C, the planktonic cells were removed from each well and the plate was rinsed with physiological saline. Then, the plate was left to dry for 10 min in a laminar-flow box. After drying, 200 µL of the corresponding sterile medium containing the phenolic compounds at a fixed concentration of 500 µg/mL was added to each treated well. For positive controls, 200 µL of growth medium was added to the treated wells, while the negative controls contained the phenolic compounds in the growth medium. Plates were then incubated for 24 h at 30 °C and the biofilm cells were detected by crystal violet staining. The experiments were performed in at least two biological parallels, and six technical parallel measurements were made each time.

### 2.5. Adhesion Inhibition Assay

The most active phenolics against the yeast’s biofilm formation were tested for their antiadhesion properties on polystyrene surface using a microtiter plate-based assay [34]. A volume of 200 µL of cell suspension (approximately 10^8^ CFU/mL), which was prepared with the corresponding growth medium and contained 500 µg/mL of the phenolic compound, was transferred into the wells of a 96-well polystyrene microtiter plate (Sarstedt, Nümbrecht, Germany). Inoculated growth medium without addition of phenolic compounds was considered as the positive control, while negative controls contained phenolic compounds in the growth medium. After setting up the culturing environments, the plates were incubated for 4 h at 30 °C, then the adhered cells were detected by crystal violet staining. The experiments were repeated at least twice, with six parallel measurements performed after the staining.

### 2.6. Crystal Violet Staining

Biofilms and adhered cells obtained in the presence or absence of phenolics were detected by the crystal violet staining method. After the incubations, supernatants were removed, and the wells were rinsed with physiological saline. To fix the biofilms and the adhered cells, 200 µL methanol was added to each well and the plates were incubated for 15 min at room temperature. After the methanol had been removed, a volume of 200 µL of 0.1% (*w*/*v*) crystal violet solution was added to each well and the plates were incubated for 20 min at room temperature. The excess dye was then removed by washing the plates under slow running tap water. The bounded crystal violet dye was released by adding 200 µL of 33% (*v*/*v*) acetic acid and incubating the plates for 10 min at room temperature. The absorbance was then measured at 590 nm (SPECTROstar Nano microplate reader, BMG Labtech, Offenburg, Germany), and the percentage of biofilm formation was calculated. Optical density detected in the corresponding positive control sample was considered as 100%.

### 2.7. Fluorescence Microscopy Studies

Fluorescence microscopy was carried out to visualize the yeast biofilms formed in the presence and absence of selected phenolic compounds. In this experiment, attaching surface for the cells was a sterilized glass microscope slide (26 × 76 mm) placed in the center of a sterile Petri dish (diameter of 90 mm). Yeasts cells were grown for 24 h at 30 °C in medium appropriate for their growth. After a homogenization step by a gentle vortexing, a volume of 6 mL from the cell suspension (approximately 10^8^ CFU/mL) was supplemented with phenolic compound to be tested reaching a final concentration of 500 µg/mL. Then, the phenolic compound contained suspension was transferred to the dish covering the surface of the glass slide. A glass surface treated with phenolic-free cell suspension was taken as control. The Petri dishes were then incubated at 30 °C for 24 h, and the biofilms formed were stained with 20 µL of 1% (*w*/*v*) acridine orange dye (Sigma–Aldrich, Munich, Germany). The excess stain was washed out by distilled water and the dyed glass slides were visualized with AxioLab (Carl Zeiss, Oberkochen, Germany) fluorescence microscope equipped with an Axiocam 503 mono (Carl Zeiss, Oberkochen, Germany) camera. Excitation and emission wavelengths of 500 and 526 nm, respectively, were used to examine the acridine orange staining.

### 2.8. Statistical Analysis

Assays were performed in at least three independent experiments and data were expressed as means ± standard deviation. Basic statistical analysis of data, such as calculation of means and standard deviations, was conducted using Microsoft Office Excel 2016 function. Significance was calculated by one-way analysis of variance (ANOVA) followed by Tukey’s multiple comparison test in the GraphPad Prism 6.00 software (GraphPad Software Inc., San Diego, CA, USA). A *p*-value of <0.05 was considered as statistically significant.

## 3. Results

### 3.1. Influence of Phenolic Compounds on Yeast Growth

A total of 15 phenolic compounds belonging to different classes (Table 1) were analyzed for their ability to inhibit the planktonic growth of four yeast strains, namely *P. anomala*, *S. cerevisiae*, *S. pombe* and *D. hansenii* through a broth microdilution method at 30 °C. In these tests, growth of yeasts was studied in the presence of phenolics at concentrations ranging from 15.625 to 2 mg/mL. The results for MIC of phenolic compound are summarized in Table 2. For most phenolic compounds, no inhibitory effect greater than 90% was identified even at the highest concentration (2 mg/mL) used. MIC was obtained at 2 mg/mL concentration for vanillin, *p*-coumaric acid and ferulic acid against *S. cerevisiae* and *S. pombe*. Quercetin, (−)-epicatechin, resveratrol and 4-hydroxybenzaldehyde resulted in MIC also at 2 mg/mL for *S. cerevisiae*, *S. pombe*, *D. hansenii* and *P. anomala*, respectively. MIC of 1 mg/mL was identified for vanillin against *D. hansenii* and *P. anomala*. Cinnamic acid exhibited MIC at 500 µg/mL concentration against all yeasts studied (Table 2). Taken together, the studied yeasts were quite sensitive to cinnamic acid and vanillin.

Apart from that most phenolics had no MIC value up to 2 mg/mL, there were noticeable inhibitions within the tested concentration range. For instance, all compounds significantly inhibited the growth of the tested yeasts at 1 mg/mL (*p* < 0.05) (Appendix A). In addition, a considerable growth inhibitory effect of the tested phenolics was recorded at a concentration of 500 µg/mL, except for (−)-epicatechin and resveratrol against *S. cerevisiae* (Appendix A).

The growth of *S. cerevisiae* was significantly inhibited by cinnamic acid even at lower concentrations than its MIC value, having more than 70% growth inhibition at 250 µg/mL (*p* < 0.05) (Appendix A). Polydatin was also effective against *S. cerevisiae* presenting higher growth inhibition than 45% at 1 mg/mL concentration (*p* < 0.05) (Appendix A). For the other phenolic compounds tested, lower inhibitory effects of less than 45% were observed even at a concentration of 1 mg/mL in *S. cerevisiae*. Among these phenolics, resveratrol exhibited the lowest inhibitory activity having less than 10% growth inhibition at 1 mg/mL (*p* < 0.05) (Appendix A). Overall, *S. cerevisiae* had notably high resistance against the growth inhibition activity of most of the phenolic compounds compared to the other tested yeast strains.

The planktonic growth of *S. pombe* was highly suppressed by cinnamic acid presenting more than 60% growth inhibition at a concentration of 250 µg/mL (*p* < 0.05) (Appendix A). Vanillin was highly active against *S. pombe* at 1 mg/mL with an 80% growth inhibition effect (*p* < 0.05), that was reduced to 32% and 13% at concentrations of 500 µg/mL and 250 µg/mL, respectively. Polydatin and 4-hydroxybenzaldehyde demonstrated more than 70% growth inhibition at 1 mg/mL (*p* < 0.05) (Appendix A). Moreover, a 26% inhibition of *S. pombe* growth was detected for the polydatin at a concentration of 250 µg/mL (Appendix A).

The planktonic growth of *D. hansenii* was significantly inhibited by cinnamic acid, vanillin and 4-hydroxybenzaldehyde even at a concentration of 250 µg/mL exhibiting inhibitory effects of 92%, 66% and 50%, respectively (*p* < 0.05) (Appendix A). At 500 µg/mL, all the phenolic compounds had significant growth inhibition against *D. hansenii* (Appendix A). At 1 mg/mL, most of the phenolic compounds had moderate to high growth inhibition against *D. hansenii*, with polydatin having the least growth inhibition at 17% (Appendix A).

*P. anomala* was sensitive to low concentrations of cinnamic acid with 80% growth inhibition at 250 µg/mL (Appendix A). Vanillin was also highly active against *P. anomala* having about 40% growth inhibition both at concentrations of 500 and 250 µg/mL (Appendix A and Appendix A). 4-Hydroxybenzaldehyde and *p*-coumaric acid had more than 80% and 70% growth inhibition, respectively, at 1 mg/mL against *P. anomala* (Appendix A).

### 3.2. Influence of Phenolic Compounds on the Formation of Yeast Biofilms

In this assay, inhibitory activity of phenolic compounds on the biofilm formation of the studied spoilage yeasts was investigated at a fixed concentration of 500 µg/mL. Vanillin was an effective antibiofilm agent against *D. hansenii* with a 97% inhibition effect (Figure 1A). Vanillin was highly active against the biofilm formation of *P. anomala* and *S. pombe* as well, with inhibition effects of 77% and 82%, respectively (Figure 1B,C). In *S. cerevisiae*, percent inhibition of biofilm growth in the presence of vanillin was below 50% (Figure 1D). These results demonstrated that the biofilm inhibitory activity of the phenolic compounds differed depending on the studied spoilage yeast. Cinnamic acid proved to be a potent antibiofilm agent against *D. hansenii*, *P. anomala* and *S. pombe* with more than 80% biofilm formation inhibition at the studied concentration (500 µg/mL) (*p* < 0.05) (Figure 1A–C). In *S. cerevisiae*, 4-hydroxybenzaldehyde was the most active phenolic compound with a 64% biofilm formation inhibition (*p* < 0.05) (Figure 1D). The *S. cerevisiae* biofilm was notably resistant against the activity of most phenolic compounds. Among the tested phenolics, only (‒)-epicatechin and 4-hydroxybenzaldehyde had more than 50% biofilm formation inhibition in *S. cerevisiae* (*p* < 0.05). It is worth mentioning that ferulic acid and protocatechuic acid had a slight stimulative effect on biofilm formation of *S. cerevisiae* (Figure 1D). Nevertheless, ten phenolic compounds, i.e., vanillin, gallic acid, (−)-epicatechin, polydatin, resveratrol, cinnamic acid, syringic acid, *p*-coumaric acid, ferulic acid and 4-hydroxybenzaldehyde, exhibited high antibiofilm activity against the osmotolerant and halotolerant spoilage yeast *D. hansenii*, with more than 50% inhibitory effect (Figure 1A).

Colonization and biofilm formation of the studied yeasts in the presence or absence of phenolic compounds were also investigated on the glass surface of a microscope slide. After 24 h incubation, the biofilms formed were visualized by fluorescence microscopy. Since vanillin, cinnamic acid and (−)-epicatechin presented high antibiofilm activities in the microdilution experiments, these compounds were selected for the assay. It was observed that samples containing the corresponding phenolic compound resulted in fragmented biofilms when compared to the control, which had an intact mature biofilm (Figure 2). Anyway, formation of all yeast biofilms on glass surface was blocked by the phenolics at the studied concentration (500 µg/mL). However, further studies were considered necessary to analyze whether the compounds could inhibit only the biofilm formation, or they also could affect the cell adhesion during incubations.

### 3.3. Influence of Phenolic Compounds on Yeast Adhesion

The activity of phenolic compounds was also tested against the adhesion of yeasts to polystyrene surface. For these experiments, the phenolics were selected based on their effectiveness as biofilm formation inhibition agents in the relevant yeast. Besides vanillin and cinnamic acid examined on each yeast, the cultivation systems contained polydatin and syringic acid, *p*-coumaric acid and 4-hydroxybenzoic acid, ferulic acid and 4-hydroxybenzaldehyde and (‒)-epicatechin and 4-hydroxybenzaldehyde for *D. hansenii*, *P. anomala*, *S. pombe* and *S. cerevisiae*, respectively. As seen in Figure 3, all phenolic compounds were significantly effective against the 4-h adhesion of yeast cells tested (*p* < 0.05). In *D. hansenii*, for instance, cinnamic acid and vanillin resulted in more than 50% adhesion inhibition (*p* < 0.05) (Figure 3A). For *P. anomala*, vanillin, cinnamic acid and *p*-coumaric acid were more effective than the 4-hydroxybenzoic acid; however, more than half of the cell mass in the sample were able to adhere in this environment compared to the control (Figure 3B). Cinnamic acid and vanillin were also highly active against *S. pombe* with adhesion inhibitions of 91% and 81%, respectively (*p* < 0.05) (Figure 3C). *S. cerevisiae* was notably resistant against the adhesion inhibition activity of the assayed phenolics compared to the other yeast strains. The cinnamic acid was the most active phenolic compound against *S. cerevisiae* with an adhesion inhibition of 51%, while the other phenolic compounds tested exhibited less than 25% antiadhesion activity (Figure 3D).

## 4. Discussion

Many foods and food additives can be an ideal substrate for the proliferation of spoilage yeasts and other microbial food contaminants. The use of antimicrobial agents to preserve food is, therefore, crucial to mitigate economic losses caused by food deterioration [2]. Chemical preservatives have been used traditionally in food preservation. However, concerns on their safety and the increasing microbial resistance towards their action led researchers to look for more effective and ecofriendly alternatives. Biofilm formation in food spoilage yeasts has only exacerbated the challenge of food preservation since biofilm structures are more resistant to antimicrobial agents than planktonic cells [9]. Many phenolic compounds have been identified as antimicrobials in previous studies [19], however, there are less data on their action towards the growth and biofilms of spoilage yeasts. Present study dealt with the potential inhibitory properties of hydroxybenzoic acid (gallic, vanillic, syringic, 4-hydroxybenzoic and protocatechuic acids), hydroxycinnamic acid (cinnamic, ferulic, *p*-coumaric and caffeic acids), stilbene (resveratrol and polydatin), flavonoid ((−)-epicatechin and quercetin) and phenolic aldehyde (vanillin and 4-hydroxybenzaldehyde) phenolics on planktonic and biofilm growth, as well as on the adhesion of *D. hansenii*, *P. anomala*, *S. pombe* and *S. cerevisiae*. These yeasts can cause deterioration of dairy products, wines, fruit juices or non-carbonated soft drinks [35]. In addition, some of them can generate unpleasant odors or flavors from preservatives frequently used in wines or fruit juices [36]. *P. anomala* is a well-known biofilm forming yeast on the surfaces of pickled vegetables [35]. Results revealed that many tested compounds had considerable inhibitory effect on the abovementioned growth activities of the food spoilage yeasts involved.

Phenolics belonging to the same group generally exhibited varying inhibitory effects on the planktonic and biofilm growth of food spoilage yeasts tested, which may be attributed to the differences in their chemical nature [37]. Among the hydroxycinnamic acids, for instance, cinnamic acid that has an acrylic acid group substituted on the phenyl ring had more robust effect on the yeasts than *p*-coumaric acid, caffeic acid and ferulic acid that had hydroxy and methoxy groups substituted on the phenyl ring. Accordingly, it is assumed that the type of the functional group attached to the phenyl ring can contribute to the inhibitory effect of the phenolic compounds exerted against the studied yeasts. This observation agrees with an earlier report indicating that the functional side groups in phenolics could have a role in the inhibition of yeast growth [37]. In addition, the high anion toxicity of cinnamic acid due to its high hydrophobicity can cause growth inhibition in *S. cerevisiae* [38]. Cinnamic acid and its derivatives can also inhibit microbial enzymes. For instance, cinnamic acid is a competitive inhibitor of benzoate 4-hydroxylase, which is an important enzyme in the degradation of aromatic compounds by fungi [39]. Furthermore, cinnamates can inhibit the activity of the β-ketoacyl acyl carrier protein reductase enzyme of the fatty-acid biosynthetic pathway [40]. Quercetin moderately inhibited both the planktonic and the biofilm growth of most yeast strains involved compared to other phenolics tested. In this context, it is worth noting that quercetin can be protective in yeasts against stress conditions by modulating signaling pathways, such as those that are involved in carbohydrate metabolism and cell wall biogenesis [41].

The mechanism of action of phenolics against microbial targets could ascribe to several effects including destabilization of the plasma membrane, modification of cellular permeability, inhibition of microbial enzymes and inactivation of efflux pumps [30,42]. Kumari et al. [43] reported that phenolics are capable of disrupting biofilm cell surface leading to an elevation in cell roughness and reduction in cell height. In a molecular docking study, Perez-Castillo et al. [44] reported the antifungal activity of cinnamic acid derivatives against *Candida* strains and identified that the compounds can bind to multiple targets. In *Candida*, caffeic acid has also been shown to be an effective inhibitor [45]. In addition, caffeic acid derivatives can interfere with the 1,3-β-glucan synthase enzyme important in fungal cell wall synthesis [46].

Among the yeasts involved in this study, *S. cerevisiae* was exceptionally tolerant to the phenolic compounds. This may be attributed to the expression of inhibitor-resistant transporters, the unique ability to convert phenolics into inactive products, and the production of extracellular polymeric substances that can act as a barrier to the phenolic inhibitors [13,28]. Investigation of phenolic tolerance mechanisms in yeasts is a growing research area nowadays. In such studies, the use of functional genomics tools, such as chemogenomic screens will help researchers to better understand the key factors associated with the phenolics tolerance in food spoilage yeasts [47].

Vanillin and cinnamic acid considerably impaired 4-h yeast adhesion at the studied concentration (500 µg/mL). However, it is worth pointing out that cinnamic acid at a concentration of 500 mg/mL exerted a strong inhibitory effect on the planktonic growth of yeasts affecting the activity of cells to be adhered. Further, polydatin, syringic acid, *p*-coumaric acid, 4-hydroxybenzoic acid, ferulic acid, 4-hydroxybenzaldehyde and (‒)-epicatechin also prevented adhesion in certain yeasts compared to the phenolic-free control. Microbial adhesion is facilitated by adhesins, which are cell wall glycoproteins that promote binding to abiotic surfaces and cell-cell adhesion [48,49,50]. Phenolic compounds can reduce the adhesion ability of yeasts by various mechanisms, such as inhibition of biosynthesis of cell wall components, interference with localization of glycosylphosphatidylinositol (GPI)-anchored proteins, and downregulation of genes that encode adhesins [51]. Adhesion inhibitory activity of phenolic compounds has been demonstrated in previous studies as well. For instance, anti-adhesion activity against *Candida albicans* were registered for the hydrolysable tannin fraction of Mangrove in the study of Glasenapp et al. [52]. Curcumin, magnolol and honokiol phenolics considerably prevented adhesion in *C. albicans* through downregulation of genes encoding adhesins [53,54].

Finally, by acting as stressors, phenolic compounds can increase the adhesion and biofilm formation of microorganisms [27]. In our studies, adhesion of yeast cells was not affected positively by the presence of phenolics. Concerning biofilm formation, however, ferulic acid and protocatechuic acid caused a slight stimulation in *S. cerevisiae*.

## 5. Conclusions

This study presented the antimicrobial, antibiofilm and antiadhesion effects of phenolic acid, flavonoid, stilbene and phenolic aldehyde compounds against *D. hansenii*, *P. anomala*, *S. pombe* and *S. cerevisiae* food spoilage yeasts. The results showed considerable inhibitory effect for many tested phenolics. However, a high variation in antimicrobial activity of phenolic molecules was detected, that may be deduced to the nature of the functional group attached to the phenyl ring. Among the analyzed phenolics, cinnamic acid and vanillin emerged as the most potent compounds against the planktonic and biofilm growth, as well as the adherence of tested yeasts. These bioactive compounds have been given less attention as potential antimicrobial agents against food spoilage yeasts, although they are considered as health-promoting natural compounds. Concerning yeasts studied, growth activities of *S. cerevisiae* proved to be the most tolerant towards many phenolics that may be due to the improved phenolic tolerance mechanisms of this fungus. In conclusion, the findings of this study could lead to the development of novel molecules active against food spoilage yeasts. The compounds identified as effective anti-yeast molecules can be used as natural food preservatives and/or sanitizers to control food-contaminating yeasts.

## Figures and Tables

**Figure 1 foods-10-01652-f001:**
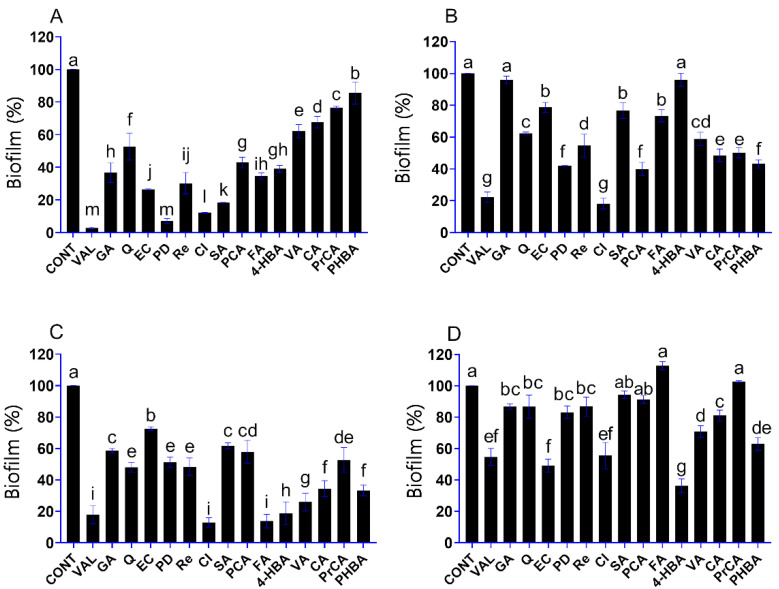
Effect of phenolic compounds (in 500 µg/mL) on the biofilm formation of *D. hansenii* SZMC 8045Mo (**A**), *P. anomala* SZMC 8061Mo (**B**), *S. pombe* SZMC 1280 (**C**) and *S. cerevisiae* SZMC 1279 (**D**). Phenolic compounds: vanillin, VAL; gallic acid, GA; quercetin, Q; (‒)-epicatechin, EC; polydatin, PD; resveratrol, Re; cinnamic acid, CI; syringic acid, SA; *p*-coumaric acid, PCA; ferulic acid, FA; 4-hydroxybenzaldehyde, 4-HBA; vanillic acid, VA; caffeic acid, CA; protocatechuic acid, PrCA; 4-hydroxybenzoic acid, PHBA. The control (CONT) represents the biofilm formation in the absence of phenolic compounds. The results shown are the mean percent biofilm formed relative to control biofilm; error bars represent standard deviation. The different letters above the columns indicate significant differences according to one-way ANOVA followed by Tukey’s multiple comparison test (*p* < 0.05).

**Figure 2 foods-10-01652-f002:**
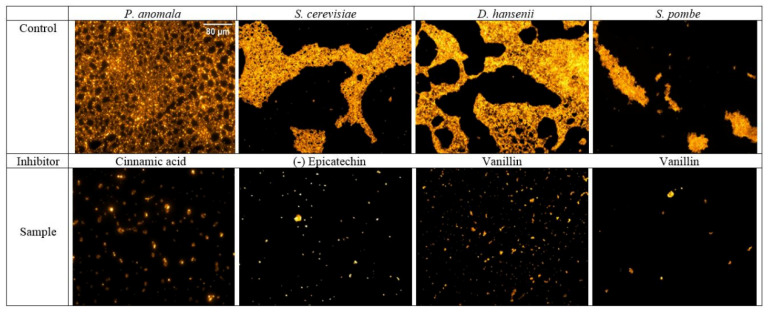
Biofilm formation of yeasts on a glass surface in the presence (sample) or absence (control) of selected phenolic compounds. Phenolic compounds were used in the samples at a concentration of 500 µg/mL. After an incubation at 30 °C for 24 h biofilms were stained with acridine orange and examined by fluorescence microscope. Magnification 20×.

**Figure 3 foods-10-01652-f003:**
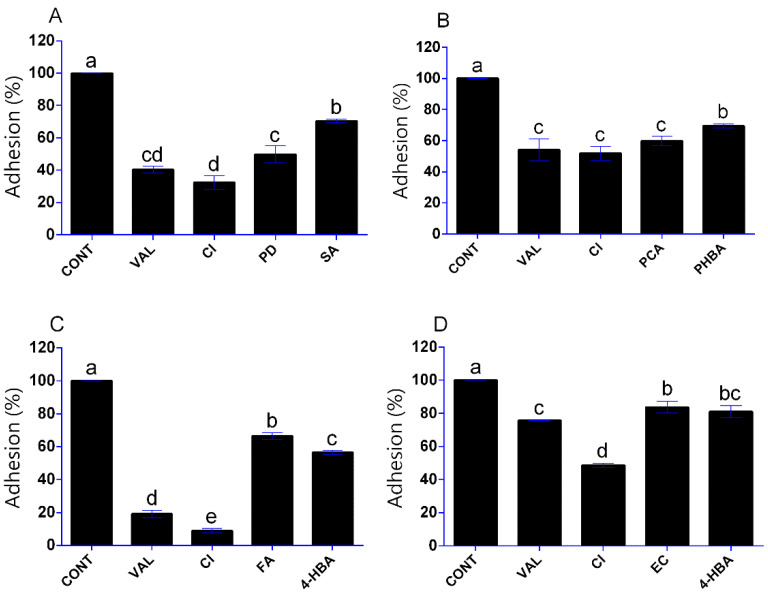
Effect of selected phenolic compounds (in 500 µg/mL concentration) on the adhesion of *D. hansenii* SZMC 8045Mo (**A**), *P. anomala* SZMC 8061Mo (**B**), *S. pombe* SZMC 1280 (**C**) and *S. cerevisiae* SZMC 1279 (**D**) yeasts to polystyrene surface. Phenolic compounds: vanillin, VAL; cinnamic acid, CI; polydatin, PD; syringic acid, SA; *p*-coumaric acid, PCA; 4-hydroxybenzoic acid, PHBA; ferulic acid, FA; 4-hydroxybenzaldehyde, 4-HBA; (‒)-epicatechin, EC. Adhesion of cells in the absence of phenolic compounds was taken as 100% (CONT). Results are presented as mean of replicates; error bars represent standard deviation. The different letters above the columns indicate significant differences according to one-way ANOVA followed by Tukey’s multiple comparison test (*p* < 0.05).

**Table 1 foods-10-01652-t001:** Phenolic compounds used in the study.

Chemical Group	Compound	Chemical Structure
Hydroxybenzoic acids	Vanillic acid	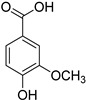
Syringic acid	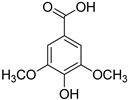
Gallic acid	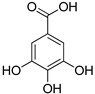
Protocatechuic acid	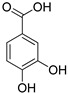
4-Hydroxybenzoic acid	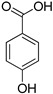
Hydroxycinnamic acids	Cinnamic acid	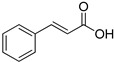
*p*-Coumaric acid	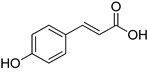
Caffeic acid	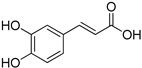
Ferulic acid	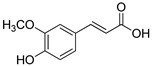
Stilbenes	Resveratrol	
Polydatin	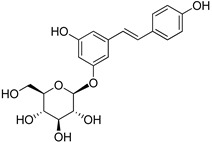
Flavonoids	(−)-Epicatechin	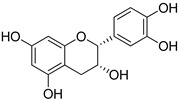
Quercetin	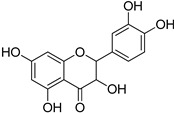
Phenolic aldehydes	Vanillin	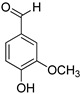
4-Hydroxybenzaldehyde	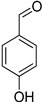

**Table 2 foods-10-01652-t002:** Minimum Inhibitory Concentration (MIC) of phenolic compounds against food-contaminating yeasts.

Phenolic Compounds	MIC (mg/mL)			
	*S. cerevisiae*	*S. pombe*	*D. hansenii*	*P. anomala*
Vanillic acid	>2	>2	>2	>2
Syringic acid	>2	>2	>2	>2
Gallic acid	>2	>2	>2	>2
Protocatechuic acid	>2	>2	>2	>2
4-Hydroxybenzoic acid	>2	>2	>2	>2
Cinnamic acid	0.5	0.5	0.5	0.5
*p*-Coumaric acid	2	2	>2	>2
Caffeic acid	>2	>2	>2	>2
Ferulic acid	2	2	>2	>2
Resveratrol	>2	>2	2	>2
Polydatin	>2	>2	>2	>2
(−)-Epicatechin	>2	2	>2	>2
Vanillin	2	2	1	1
4-Hydroxybenzaldehyde	>2	>2	>2	2
Quercetin	2	>2	>2	>2

## Data Availability

Data are contained within the article and Appendix A.

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
