# Peer review of "In Vitro Activity of Selected Phenolic Compounds against Planktonic and Biofilm Cells of Food-Contaminating Yeasts"

_foods, 2021, doi:10.3390/foods10071652_

Round 1
Reviewer 1 Report
The presented manuscript is well designed and written.
The results are interesting and cleraly presented, however I have one small question to be explained:
- What was the key for choosing the polyphenos for the study?
- Why these particular copmpounds were chosen?
After this explanation I do feel that the manuscript can be considered.
Reviewer 2 Report
The manuscript entitled “In vitro activity of selected phenolic compounds against plank- 2 tonic and biofilm cells of food-contaminating yeasts” report about inhibitory effects of various phenolic compounds on different yeast strain. All the experiments designed well, and manuscript written well with proper citation of previous work. Manuscript needs following changes for further improvement.
Comments:
Line 35: Yeasts are renowned…mention few products where yeast is used.
Line 54: growing concern against chemical additives….Add few points here regarding main concern.
Line 55: Phenolic compounds are secondary metabolites…What are different phenolics compounds and their plant source. Add more information.
Line 59: What are different bioactive molecules reported for antibiofilm activity against yeast.
